# Detection of Gait Events Using Ear-Worn IMUs During Functional Movement Tasks

**DOI:** 10.3390/s25123629

**Published:** 2025-06-09

**Authors:** Terry Fawden, Iwan Vaughan Roberts, Sarah Goldin, Yash Sharma, Henry Dunne, Thomas Stone, Manohar Bance

**Affiliations:** 1Department of Clinical Neurosciences, University of Cambridge, Cambridge CB2 0QQ, UK; ivr22@medschl.cam.ac.uk (I.V.R.); yash@rogers.com (Y.S.); henry.dunne@nhs.net (H.D.); mlb59@cam.ac.uk (M.B.); 2Department of Engineering, University of Cambridge, Cambridge CB2 1PZ, UK; sfg39@cam.ac.uk; 3Cambridge University Hospitals NHS Foundation Trust, Cambridge CB2 0QQ, UK; thomas.stone@nhs.net

**Keywords:** earables, gait event detection, inertial sensor, temporal parameters

## Abstract

Complex walking tasks such as turning or walking with head movements are frequently used to assess dysfunction in an individual’s vestibular, nervous and musculoskeletal systems. Compared to other methods, wearable inertial measurement units (IMUs) allow quantitative analysis of these tasks in less restricted settings, allowing for a more scalable clinical measurement tool with better ecological validity. This study investigates the use of ear-worn IMUs to identify gait events during complex walking tasks, having collected data on 68 participants with a diverse range of ages and movement-related conditions. The performance of an existing gait event detection algorithm was compared with a new one designed to be more robust to lateral head movements. Our analysis suggests that while both algorithms achieve high initial contact sensitivity across all walking tasks, our new algorithm attains higher terminal contact sensitivity for turning and walking with horizontal head turns, resulting in more accurate estimates of stance and swing times. This provides scope to enable more detailed assessment of complex walking tasks during clinical testing and in daily life settings.

## 1. Introduction

Human gait requires the complex interaction of the neurological, musculoskeletal and vestibular systems, making it a rich source of information for monitoring and assessment of their function. Common functional movement tests such as the Functional Gait Assessment and Berg Balance Scale, which are used to assess an individual’s postural stability and ability to perform multiple motor tasks, contain a significant number of gait-related activities [1]. Compared to straight-line walking, these provide additional challenges to standard gait, for example by adding directed head movements and following a turning route. These reflect scenarios which people may find difficult in daily life, particularly when they have an underlying condition affecting movement. While there are multiple existing methods of instrumenting and analysing these tests using camera-based methods or often simply under direct clinical supervision, recent work has moved towards monitoring and assessing gait function using wearable sensors.

Wearable sensors have been deployed in a multitude of clinical and academic studies both to study movement directly and as an addition to other sensing modalities [2,3,4,5], as well as for larger-scale population behaviour [6]. They are convenient for gait analysis since their use is not restricted to a specific location and they do not require the direct supervision of a clinician, while they require less effort to set up than a camera-based system. However, a wearable sensor must have a suitable form factor and be placed easily and consistently without interfering with a user’s natural gait pattern.

Wearable devices are often composed of inertial measurement units (IMUs), containing accelerometers, gyroscopes and sometimes magnetometers, and worn at a wide range of locations such as the feet, shank, trunk or wrist [7,8,9,10]. Each possible location provides a trade-off between wearability and availability of information about the aspect of gait being monitored, and this has been explored extensively in the previous literature [11,12]. IMU sensors placed at the ear, often referred to as ’earables’, provide a suitable solution given the constraints since the ear is a stable, comfortable and aesthetically acceptable place to house a sensor, which is reflected in widespread commercial use of headphones and earbuds [13,14,15]. Many individuals use hearing aids, especially those in ageing populations where movement disorders are more prevalent, and some commercially available earbuds and hearing aids now contain an IMU or accelerometer. Since the vestibular system is located in the inner ear, it is a useful body segment to track in itself when monitoring individuals with balance disorders. Further, due to the direct coupling to the trunk, the signals produced from a head or ear-worn sensor describe the movement of centre of mass well [16,17] and even filter out some lower body motion artifacts to produce a clearer signal for gait event detection [18]. The head’s location on the sagittal plane also means that the movement of both left and right sides of the body can be measured using the same system.

Gait analysis using IMUs involves processing of the signals produced by the accelerometer and gyroscope. While IMUs have been used to analyse many aspects of gait, most metrics used for gait analysis firstly require separation of the time series signal into gait cycles (consecutive initial ground contacts of the same foot) and subphases; hence, it is important that this is performed accurately. Further, the gait events themselves are used to calculate temporal parameters such as stance and swing times, which are clinically useful in their own right. There is a body of existing work on classifying gait events using head- and ear-worn IMUs. Jarchi et al. proposed a method based on singular spectrum analysis (SSA) with longest common subsequence (LCSS) on the anterior–posterior (AP) and mediolateral (ML) acceleration signals to determine initial contact (IC) and terminal contact (TC) events in level, downstairs and upstairs walking [19]. Jarchi’s method, however, requires averaging over many cycles and was shown to perform poorly during other studies [20]. Diao et al. applied a similar method using SSA to determine these events for both healthy and postoperative individuals during straight-line walking, but used the superior–inferior (SI) signal rather than the AP signal to identify ICs, achieving low error in stride, swing and stance time relative to a commercial IMU-based gait evaluation system [21]. Seifer et al. improved on Diao’s algorithm by using the first dominant oscillation of the ML acceleration, which reduced the error in determining the laterality of each step [20].

Xu et al. used a deep learning approach to account for short walking bouts; however, their method suffers from high false positive rate and was tested with only thirteen participants in a non-natural walking scenario [22]. Also applying a deep learning approach, Decker et al. recently achieved an F1 score of 99% for IC detection on level ground walking across a range of walking speeds; however, they only collected data from healthy individuals and scored relatively lower (91%) for TC events [23]. Jung et al. validated five walking and six running gait parameters detected by an ear-worn system against a motion capture system and instrumented treadmill, though they did not develop a new gait parameter estimation algorithm [24]. Hwang et al. noticed a pattern of peaks in the SI signal mapping to foot-off, producing a method to detect ICs and TCs from only the SI acceleration. However, this was only tested on seven young, healthy participants and made a large assumption about the number of peaks per cycle, which varies depending on the participant’s natural gait pattern and the nature of the surface being walked on [25].

While several of these methods have been shown to perform well in straight-line walking, none have been tested in more challenging scenarios such as during turning and while walking with head turns. These have been shown to uncover underlying movement-related deficiencies which do not appear during straight-line walking and are therefore powerful clinical tools [7,26,27]. Head-worn IMUs have been used to investigate the coupling of head and trunk during turning, though not for gait event detection [28]. Other studies have investigated gait event detection during turning movements with IMUs; however, these were placed at the shank [7], feet [8] and lower trunk [27]. Further, the majority of studies in this field lack diversity in age, gender and level of movement function, with a bias towards young, typically functioning males. Movement patterns vary greatly across these cohorts, making it crucial to collect data from a representative participant group.

Developing algorithms for an ear-worn IMU-based system which are robust to turning and walking with directed head turns provides a means to quantitatively analyse these tasks in less restrictive settings. This provides a basis to develop a more scalable system to improve access to clinical tests which include these movements. Therefore, this study seeks to provide the following contributions:Evaluation of gait event detection using an ear-worn IMU system during complex gait movement involving head turns against an optical ground truth.Comparison of ear-worn IMU gait detection algorithm performance for a large participant cohort, involving both typically functioning individuals and those with an underlying movement disorder.Proposal and evaluation of a new gait event detection algorithm, TP-EAR (Temporal Parameters from the EAR), which provides more robust TC detection and improved estimation of stance and swing time during complex gait tasks.

## 2. Materials and Methods

### 2.1. Data Collection

This study collected data from 68 participants, of whom 18 had an underlying condition affecting gait and were classified as ‘Non-Typical’. Of these, 10 had Parkinson’s Disease, 3 had recent vestibular schwannoma surgery, 1 had benign paroxysmal positional vertigo, 1 had Ménière’s Disease and 3 had osteoarthritis. More details are shown in Table 1. Data was collected on an 8 m long walkway in the Cambridge Movement Laboratory.

The participants wore a bespoke headset containing 6-axis LSM6DSOx IMUs placed behind the left and right ears (shown in Figure 1), with other sensors placed at the chest, thigh, shank and wrist. For consistency, we analysed data from the left ear sensor; however, we found that the right ear produced generally consistent results compared to those from the left ear and these are shown in Appendix A. The sample rate was set to 100 Hz. All IMUs were connected to an ESP32 Thing Plus development board via a multiplexer which was attached to the chest strap and powered by a 180 mAh lithium polymer battery. This sent the data to a laptop in real time via WiFi to be processed offline.

Sixteen optical markers were placed on each participant following the lower body plug-in gait model. These were captured by the optical motion capture system (OMC) (VICON, Oxford, UK) with 12 infrared cameras and two RGB cameras placed perpendicularly and in parallel to the walkway. The ground truth gait events were labelled manually using the video footage obtained from the RGB cameras, except when they were obstructed or the participant went out of view, in which case the markers were used as described in [29]. The IMUs and camera system were triggered at the same time, and any small differences in start time were corrected by maximising the cross-correlation between them. The participants then completed the following series of tasks, which are illustrated in Figure 2:Walk: Walking in a straight line down the walkway. Each participant completed as many repetitions it took for 3 clean force plate strikes per foot; otherwise, no more than 12 repetitions.WalkV: Walking in a straight line down the walkway while performing vertical head turns. Each participant completed 3 repetitions.WalkH: Walking in a straight line down the walkway while performing horizontal head turns. Each participant completed 3 repetitions.Turn: The participant performed a timed up-and-go task, which involved getting up from a chair, walking up to and turning around a cone positioned 3 m away from the chair, then returning and ending in a seated position. This study considers only the time between the first and last steps, with the turning period identified from the IMU signal and analysed independently (detailed in Section 2.3). Each participant completed 3 repetitions.

**Figure 2 sensors-25-03629-f002:**
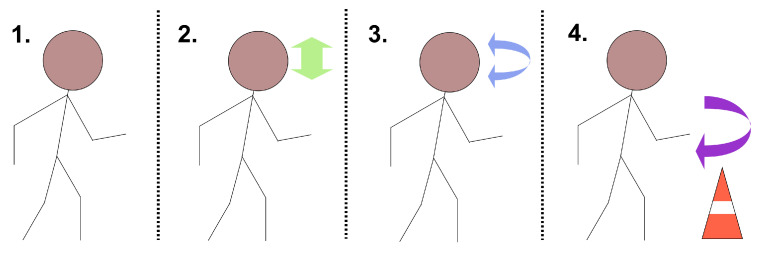
The four walking tasks, labelled left to right as 1. Walk, 2. WalkV, 3. WalkH and 4. Turn. The arrows in 2. and 3. indicate the direction of head movement and the arrow in 4. indicates the route taken by the participant around the cone while turning.

All walking tasks were completed barefoot at a self-selected speed. The head turning rate for WalkH and WalkV was instructed to be approximately 2 Hz, communicated by demonstration, and participants were instructed to complete the timed up-and-go task as quickly and as safely as possible. One non-typically functioning participant did not complete WalkH and WalkV and one typically functioning participant did not complete Turn. In 45 out of 1208 trials (3.7%), either more than 10 consecutive samples were missed from the IMU or there was a gap within 0.3 s of an IC (which would directly interfere with its detection). This occurred due to lost data packets during transmission of the IMU data from the device to the laptop. These trials were excluded from analysis to ensure there were no anomalies during the detection of gait events due to low data quality.

### 2.2. Data Preprocessing

All data analysis and preprocessing was performed in Python 3.9. Once each trial had been checked for significant gaps as detailed in Section 2.1, any remaining gaps were filled using a cubic spline. The data from each accelerometer and gyroscope axis was then rearranged to fit a north–east–down convention, since this was the system used by the Kalman filter which produced the IMU’s orientation estimate as discussed in Section 2.3. The data for each trial was saved in a .csv file and loaded using the Pandas package before being passed as input to the Diao and TP-EAR algorithms, which are described in Section 2.4.

### 2.3. Definition of Turning

The accelerometer and gyroscope data from each IMU was fused using an error-state extended Kalman filter, providing an estimate of the IMU’s orientation during the trial. This was adapted into Python from an existing Matlab implementation [30]. Each trial contained exactly one turn, which was identified from the yaw angle, ψ using the following criteria: (1)$$Start:\;|\psi |>15^{\circ }\;\mathrm{and}\;\dot{\left|\psi \right|}>30^{\circ }/s$$(2)$$End:\;|\psi |\geq {\left|\psi \right|}_{max}-5^{\circ }$$

This accounts for the fact that the head begins to turn before the body’s centre of mass [31], while the ending is reached when the head has reached its maximum rotation. The 5° and 15° offsets greatly reduce the number of false positives due to non-turning-related head movements. These thresholds empirically matched the start and ending of the turns identified using the pelvis markers on the OMC. A demonstration of the turn segmentation process is shown in Figure 3.

### 2.4. Gait Event Detection

This section describes our implementation of Diao et al.’s gait event detection algorithm [21] and our newly proposed algorithm, TP-EAR. The code may be found at https://github.com/terrylboro/GaitEventDetection/tree/master, accessed on 6 June 2025. A demonstration of both methods on a sample gait cycle is shown in Figure 4.

#### 2.4.1. Diao

This method was initially described by Diao et al. and is based on singular spectrum analysis (SSA) [21]. SSA is a nonparametric technique based on singular value decomposition in which a time series signal is decomposed into multiple underlying parts then reconstructed and grouped into trend, periodic and noise components [32,33].

Diao’s method applies SSA on the SI and ML acceleration signals. The peaks of the dominant frequency extracted from the SI signal are classed as ICs. Applying the improvement suggested by Seifer et al. [20], the laterality (i.e., which foot touched the floor) is determined using the signed difference between consecutive samples of the ML dominant frequency, with a positive result indicating a left IC and, otherwise, a right IC. The corresponding TC is then determined by finding the next local minimum or maximum in the trend-removed ML signal depending on the IC laterality.

#### 2.4.2. TP-EAR

Diao’s method relies on the consistency of the ML acceleration signal across gait cycles; however, this does not hold if the participant is looking side-to-side or performing a turning motion. Further, the accuracy of TC detection is coupled to the IC laterality, meaning that misclassification of an IC causes the algorithm to look for a minimum rather than a maximum, reducing the time accuracy of the detected TC.

While the ML signal is inherently variable during lateral head turns, the SI signal remains relatively unchanged. TC has been observed to roughly correspond to a peak in the SI acceleration signal measured at both the lower trunk and the head [10,25]. By locating this peak in the ear-worn SI acceleration signal, and taking advantage of the proven high accuracy of IC detection using SSA, we developed an improved SSA-based algorithm to detect IC and TC events using the SI signal. The process is shown in Figure 5.

Firstly, the SI signal is band-pass-filtered between 0.5 Hz and 12 Hz—the upper cut-off is higher than the 5 Hz used by the Diao algorithm but we found that this improved the visibility of the peaks corresponding to IC and TC. Then SSA is performed on the SI and ML signals in the same way as for the Diao algorithm. The peaks of the dominant SI signal are used to estimate the IC events. Isolating a ‘trusted swing’ region has been shown to improve the robustness of gait detection algorithms [11]. Therefore, for each dominant peak, a window is opened between the point before the peak at which the dominant SI signal is at 80% of the peak amplitude and the next minimum in the dominant SI signal. Inside this window, the number of peaks in the original SI acceleration signal are counted. If there are two peaks (which usually occurs in straight-line walking), the first peak is identified to be the IC event and the second to be the TC event. If there are more than two peaks, the peak closest to the dominant SI signal peak is determined to be the IC. TC is the latest SI acceleration peak which is greater than zero. Finally, if there is one peak, a peak-sharpening algorithm is applied to the signal. If this uncovers at least one more peak, these are processed the same as the aforementioned cases. Otherwise, IC is determined to be the dominant SI peak and TC is determined to be the absolute minimum gradient from this point until the end of the window. Laterality is determined using the dominant ML acceleration signal in the same way as the Diao algorithm.

### 2.5. Evaluation

The gait events were calculated using both the Diao and TP-EAR algorithms for both right ear and left ear. The SSA window length was set to 2 s, which is higher than other studies since it was found that some of the more impaired participants walked at a lower frequency and hence required a longer window length to achieve sufficient separation between reconstructed components. For a small number of trials which were shorter than 4 s, the window length was reduced to 1 s and the process repeated.

#### 2.5.1. Detection Sensitivity and Laterality Accuracy

The detected gait events were compared with the ground truth events firstly by considering all ICs and TCs together regardless of laterality. If an IMU-derived event differed by more than 300 ms from a ground truth event, this was considered as a true negative. Likewise, if for an IMU-derived event, there was no corresponding ground truth event within 300 ms, this was considered to be a false positive. This was to provide consistency between the results of this study and existing work [20,34]. For all true positive events, the laterality was compared and the accuracy was calculated as the percentage of correct matches.

#### 2.5.2. Time Difference

For each true positive event, the difference between the IMU time and ground truth was calculated. The mean and standard deviation of absolute error was then calculated across all participants. Stride, stance and swing times were calculated using Equations (Equation 3) to (Equation 5), with the IMU-derived and ground truth events provided as input. The error was calculated by subtracting the ground truth temporal parameter values from those calculated using the Diao and TP-EAR gait events. We consider both absolute error (AE)—i.e,. the absolute value of the calculated error—and signed error (SE)—i.e., the error value taking into account whether the IMU produced an over- or under-estimation.(3)$$StrideTime=\frac{IC_{end}-IC_{start}}{f_{s}}$$(4)$$StanceTime=\frac{TC-IC_{start}}{f_{s}}$$(5)$$SwingTime=\frac{IC_{end}-TC}{f_{s}}$$
where $f_{s}$ is the system sampling frequency.

## 3. Results

### 3.1. Detection Sensitivity and Laterality Accuracy

The sensitivity of IC and TC event detection was calculated for each activity as detailed in Section 2.5.1. The results for the Typical and Non-Typical participant groups are shown in Table 2 and Table 3, respectively.

#### 3.1.1. Typical Sensitivity

For both algorithms, IC sensitivity was high across all activities. The Diao algorithm attained scores of 99.9% for Walk, WalkV and WalkH, though performed slightly worse on the turn with 99.4%. TP-EAR achieved the same scores for Walk, WalkV and Turn, but was 0.1 pp lower for WalkH (99.8%).

TC sensitivity was lower than IC for all activities for both algorithms, apart from Walk Diao, which scored 100%. This was caused by a small number of IC events being missed at the very end of the walking bout, though the overall score was still 99.9%. The Diao algorithm performed well for Walk and WalkV (100% and 99.7%), though scored slightly lower for WalkH (97.1%) and much lower for Turn (91.4%). TP-EAR attained 99.9%, 100% and 99.8% sensitivity for Walk, WalkV and WalkH, though this was slightly lower for Turn (98.7%). The performance improvement for TP-EAR versus the Diao algorithm was greatest for Turn at 7.3 pp, followed by WalkH at 2.7 pp then 0.3 pp for WalkV, while it scored 0.1 pp lower for Walk.

#### 3.1.2. Non-Typical Sensitivity

Similarly to the Typical group, both algorithms achieved high IC sensitivity across all activities. The Diao algorithm scored 99.8%, 100% and 99.5% for Walk, WalkH and Turn, which were both within 0.1 pp of the same activities in the Typical group. For WalkV, Diao scored 98.5%, which was 1.4 pp lower than in the Typical group. TP-EAR followed a similar pattern, achieving high scores for Walk, WalkH and Turn (99.8%, 100% and 100%) but slightly lower for WalkV (99.6%).

The TC sensitivity scores were again lower than IC for all activities for both algorithms, apart from Walk TP-EAR, where they were the same. TP-EAR and Diao achieved similar TC sensitivity for Walk, with 99.7% and 99.8%, respectively. However, while the TP-EAR sensitivity remained comparably high for WalkV and WalkH (99.5% and 100%) compared to Walk, Diao’s sensitivity fell to 98.4% and 98.1%, which was 1.1 pp and 1.9 pp lower. This effect was accentuated for Turn, where TP-EAR attained 100% sensitivity while the Diao algorithm scored just 93.1%.

Overall, the activity with the lowest IC sensitivity was WalkV (98.5% and 99.6% for Non-Typical) followed by Turn (99.4% and 99.4% for Typical). On the other hand, there were many activities which achieved 99.9% or more IC sensitivity for both algorithms.

Analysing the TC sensitivity, the Diao algorithm’s lowest scores came for Turn across Typical and Non-Typical groups, at 91.4% and 93.1%. Conversely, while the lowest TP-EAR score came for Turn with the Typical group, TP-EAR’s highest score (at 100%) came for Turn with the Non-Typical group, tied with WalkH Non-Typical. The highest sensitivity for the Diao algorithm came for Walk at 100% for the Typical group.

#### 3.1.3. Laterality Accuracy

For both the Typical and Non-Typical groups, laterality detection was highly accurate on the Walk activity, scoring 100% and 99.7% for Diao and 99.8% and 99.7% for TP-EAR. For WalkV, Diao and TP-EAR attained 99.7% and 99.4% for Typical and a slightly lower 97.5% and 97.8% for Non-Typical. The lowest laterality accuracy came on WalkH for both algorithms, at 82.8%/95.8% for Typical/Non-Typical with the Diao algorithm and 83.0%/95.2% with TP-EAR. For Turn, the Diao algorithm and TP-EAR scored 94.6% and 93.7%, respectively, for the Typical participant group, but both scored 100% for the Non-Typical group.

### 3.2. Temporal Parameter Comparison

The mean absolute error (MAE) across all gait cycles of each activity are shown in Table 4 (Typical) and Table 5 (Non-Typical) alongside the standard deviation in AE among all cycles. The signed errors for stride and stance times are illustrated in Figure 6 and Figure 7, respectively—swing time was excluded since this can be inferred from the stride and stance times. For reference, the absolute values calculated by each IMU-based algorithm and the ground truth are included in Appendix B.

#### 3.2.1. Typical Average

Table 4 shows that the MAE in stride time is similar between the Diao and TP-EAR algorithms for the typically functioning group. For Walk, WalkV and WalkH, the MAE is 4 ms, 4 ms and 5 ms lower for TP-EAR, while Diao is 6 ms lower for Turn. For stance time, Diao and TP-EAR achieve comparable MAE (26 ms vs. 28 ms and 33 ms vs. 36 ms) on Walk and WalkV. However, for WalkH, the MAE is much lower for TP-EAR (37 ms) compared to Diao (60 ms), and similarly for Turn, TP-EAR’s MAE is 54 ms versus Diao’s 64 ms. The MAEs for swing time follow the same trend as for stance time by activity.

The MSE in stride time is close to zero for Walk, WalkV and WalkH for both algorithms. TP-EAR is almost exactly at zero (0 ms, 1 ms and 1 ms) while Diao shows a slight positive bias at 4 ms, 3 ms and 3 ms. Both algorithms show a much stronger negative bias for Turn, at −22 ms and −20 ms, respectively; however, TP-EAR is closer to zero than Diao. The MSE attained by TP-EAR for stance and swing during WalkH is low, at 0 ms and 1 ms. Aside from during Turn, TP-EAR shows similar magnitudes for stance and swing MSE while Diao has slightly greater magnitude of error for stance during Walk and WalkV and swing during WalkH. For Turn, the magnitude of signed error is 16 ms greater for swing than stance with TP-EAR; however, for Diao, this difference is just 4 ms, with stance greater than swing. All Turn values show a negative bias.

#### 3.2.2. Typical Deviation

Observing the standard deviation in AE of the Typical group shown in Table 4, it is clear that this follows a similar trend to that of the mean. For stride time, the standard deviations were relatively comparable between Walk, WalkV and WalkH (14 ms/15 ms, 14 ms/17 ms and 15 ms/17 ms) and were slightly higher for TP-EAR compared to Diao. For Turn, this was more than twice as high for both algorithms, at 36/47 ms. When taking signed differences into account, as shown in Figure 6a, the standard deviation in stride time was slightly higher overall; however, TP-EAR was slightly lower than Diao for Walk, WalkV and WalkH. For Turn, TP-EAR still produced a higher standard deviation in SE than Diao (63 ms versus 50 ms).

The standard deviation of AE for stance time was between 0.6× and 1.4× greater for both algorithms across Walk and WalkV when compared to stride time. However, for WalkH, the standard deviation for the Diao algorithm is 2.5× higher for stance time compared to stride, while for TP-EAR, it is just 1× higher. Further, the value for TP-EAR (34 ms) is just over half that of Diao (57 ms). This effect is repeated for Turn though to a lesser extent, with the standard deviation for TP-EAR being 12 ms smaller than for Diao, but remaining similar to the stride time deviation (while Diao had a 22 ms greater standard deviation for stance time versus stride time). The standard deviations for stance and swing were similar for both algorithms.

#### 3.2.3. Non-Typical Average

For the Non-Typical group, shown in Table 5, the MAE for all activities and algorithms is generally higher, though less so for Turn. The stride time difference in MAE between Diao and TP-EAR is smaller, with Diao being 3 ms and 2 ms higher for Walk and WalkV, TP-EAR being 2 ms higher for Turn, and both being the same for WalkH.

The greatest difference in MAE for stance time occurred for WalkH (55 ms for TP-EAR versus 76 ms for Diao) and was notable though slightly lower for Turn (68 ms versus 75 ms). Swing time MAE closely followed that of stance time aside from for WalkV, where it was 6 ms and 5 ms higher for Diao and TP-EAR, respectively. In all cases, the MAE in stride time was much lower than for stance and swing time.

The MSE for stride time was generally slightly more positive overall for both algorithms, including for Turn, which was still negatively biased (−18 ms and −17 ms for Diao and TP-EAR, respectively). The differences in MSE were similar to the typically functioning participants; however, this time, Diao was slightly closer to zero than TP-EAR for WalkV, at 2 ms versus 3 ms. For stance and swing times, the signed errors were generally much higher when compared to the typically functioning participants. For WalkV, the MSE in stance/swing were −40 ms/46 ms and −26 ms/29 ms for Diao and TP-EAR, respectively, which was 34 ms/37 ms and 23 ms/27 ms greater by magnitude than for the typically functioning participants.

#### 3.2.4. Non-Typical Deviation

The standard deviation in AE for stride time was between 0.6× and 1.1× larger for the non-typically functioning participants when compared to the typically functioning participants for Walk and WalkH. However, it was 2.6× and 1.7× larger for Diao and TP-EAR, respectively, during WalkV. For Turn, the stride time standard deviation was 4 ms lower for the non-typically functioning participants versus the typically functioning participants with the TP-EAR algorithm, though it was 5 ms higher when making the same comparison for the Diao algorithm.

For stance and swing time, the pattern was similar, with the standard deviation in AE for Walk and WalkH being between 0.4× and 0.9× greater for the non-typically functioning group compared to the typically functioning group. The difference for WalkV was less significant than for stride time; however, it was still 1× larger for both algorithms. For Turn, while the standard deviations remained more similar between participant groups than for the other activities, this time, Diao produced a 4 ms lower standard deviation for the non-typically functioning group while TP-EAR was 14 ms higher.

The standard deviation in SE followed a similar trend to that of AE, though with greater magnitudes. For both AE and SE, the standard deviations in stance and swing time were highly similar.

## 4. Discussion

This study investigated the performance of two algorithms for detecting gait events during complex walking tasks using ear-worn IMUs. These were tested on data collected from 68 participants across a broad range of age, sex and size, and the sample crucially included 18 participants with a known movement disorder. Temporal parameters, namely stride, stance and swing times, were calculated using the detected events and compared with a ground truth labeled using optical data.

### 4.1. Comparison by Activity

#### 4.1.1. Sensitivity

Both algorithms generally performed strongly for IC sensitivity, with half of the activity/group combinations scoring 99.9% or higher. Given that both algorithms used SSA and peak identification to find ICs, it follows that the sensitivity was generally similar between them. The scores align with those reported by [20], who reported IC sensitivity above 99% in their implementation of the Diao algorithm for straight-line walking, and [25], who reported a 0.24% MAE in step counting around an athletics track using a head IMU. For Walk and WalkH, both algorithms missed very few IC events; however, both performed relatively worse for WalkV and Turn when averaged across both participant groups. Since WalkV incorporates vertical head movements, which adds an extra component to the SI signal, it makes sense that the algorithm performed slightly worse here. Similarly, the turn provides an interruption to the straight-line gait signal, which makes it trickier for the SSA algorithm to identify the dominant oscillation. This being said, both algorithms still identified the vast majority of IC events correctly. TP-EAR did tend to achieve slightly higher sensitivity than Diao, particularly for the more challenging WalkV and Turn, which supports the use of the underlying SI signal peak rather than simply using the dominant oscillation peak. Further, both algorithms were more sensitive than the 95–97% IC precision reported by Romijnders et al.’s study, which used shank-mounted IMUs to measure gait events during turning [7]. This suggests that they would be suitable for counting steps completed during a turn, which has been shown to differentiate between mildly impaired Parkinson’s Disease patients and age-matched controls [35].

Much greater differentiation took place when comparing the TC events. TP-EAR was highly sensitive across all activities, scoring higher for all events than the 97.5% sensitivity reported by [20] during straight-line walking and comparably to the 99% precision for healthy participants reported by [36]. However, while Diao was highly sensitive for Walk and WalkV (the lower result for WalkV is caused by compounding of the missed IC events), this dropped off slightly for WalkH and more so for Turn. The high sensitivity for Walk and WalkV would be expected since there was no change to prescribed movement pattern in the ML plane. The relatively high TC sensitivity for WalkH, where the horizontal head turning directly affected the ML acceleration signal, may have been due to the fact that while the pattern was changed, there were still peaks/troughs within 300 ms of the IC which the algorithm interpreted as TC events. Therefore, the poorer performance of the Diao algorithm on this activity was reflected more by the higher stance and swing time errors. Meanwhile, the Diao algorithm’s poor performance on Turn was likely due to a combination of the unusual foot placement and the extra disruption of the ML signal due to the imbalance imposed by the turning motion, which was avoided by TP-EAR since it relied on the SI signal.

#### 4.1.2. Laterality Accuracy

The method of laterality detection was the same for both algorithms; therefore, the results were largely comparable. Small differences stemmed from the adapted IC detection algorithm used by TP-EAR, which changed the number of detected events and location of the IC in the time series signal. The main finding was the large drop-off in laterality detection accuracy for WalkH, especially for the typically functioning group. This was expected since the horizontal head movements caused the ML signal to differ significantly from that expected during straight-line walking. During turning in the typically functioning group, the sensitivity was much higher than for WalkH but lower than Walk and WalkV. Given that there is some disruption to the ML signal over the course of the turn but less so than in WalkH, it follows that the laterality should be slightly negatively impacted relative to Walk but not so relative to WalkH. However, the laterality detection during turning reached 100% in the non-typically functioning group, which may reflect differences in their turning strategy (discussed more in Section 4.3).

### 4.2. Temporal Parameters

Both algorithms produced consistent stride time errors across Walk, WalkV and WalkH, with MAEs between 10 ms and 17 ms for the typically functioning participants and between 19 ms and 37 ms for the non-typically functioning participants. This is comparable with the stride time MAE reported using the Diao improved algorithm in [20] and to Zijlstra et al.’s lower trunk approach (between 2 ms and 15 ms depending on speed) [10]. Since the sampling interval is 10 ms, this equates to between one and four samples. The TP-EAR algorithm showed lower MSE magnitudes for these activities than Diao (and in fact was nearly zero), suggesting that using the underlying SI signal peak removes a bias introduced by using the dominant signal.

The stride time MAE for Turn was greater for both algorithms, between 40 ms and 50 ms, while the MSE showed a negative bias for both participant groups. This reflected the added difficulty for the SSA-reconstructed component to capture the change in SI signal during the turn where the nature of foot contact, and therefore the location of each IC, changed. The stride time MAE was much higher than that reported by [11] using lower leg IMUs on a slalom course (between 9.4 ms and 15.2 ms depending on algorithm and location); however, this may be due to the turn in this study being much sharper and hence more disruptive to gait. The mean and range of the stride time values calculated by the IMU still align well with the ground truth (shown in Appendix B), suggesting it still adequately captures the underlying pattern.

Both the mean and standard deviation in difference between IMU and ground truth were higher for stance and swing times, which depend on TC as well as IC accuracy. This was expected since lower accuracy and precision are frequently reported for TC versus IC events, even when using the trajectories from optical markers placed on the feet [37]. The increase in error was particularly evident in WalkH, where the TC detection was more difficult due to the disruption to the ML acceleration signal. While the difference in AE and SE is notable, it is likely that this would be much greater if the missed events were taken into account, since TP-EAR missed fewer events. The MAE across the other events was comparable between the two algorithms, though the magnitude of SE was generally much lower for TP-EAR, suggesting it was much less biased.

Across all activities, there were a handful of large outliers, some of which were larger than 300 ms, which are apparent when observing Figure 6 and Figure 7. These occurred when a gait event was detected early and the following event was detected late (or vice-versa). If one or both of these events was wrongly estimated by a margin close to the cut-off (300 ms), the calculated temporal parameter could therefore be out by more than 300 ms. While this happened rarely, depending on the application scenario it may help to incorporate a threshold to exclude known outlier values, for example when calculating average temporal parameters across many trials for a given individual.

### 4.3. Participant Group Comparison

For Walk and WalkV, the sensitivity and laterality determination was slightly lower for non-typically functioning participants when compared to the typically functioning participants, and the AE and SE were also slightly higher. This was most noticeable for WalkV with the Diao algorithm, though less so with TP-EAR, which suggests that the vertical head movements contributed to an offset in the dominant SI component which could be compensated for by TP-EAR.

Contrary to what may be expected, the sensitivity of both algorithms improved for the Non-Typical group for WalkH and Turn, and TP-EAR attained 100% sensitivity for both IC and TC. For WalkH, this may be due to differences in the timing or intensity of the horizontal movements of the head while performing the task between groups, which may be supported by the fact that they spent relatively longer in stance phase compared to Walk than the typically functioning group (as shown in Table A7), and that the laterality determination accuracy is much higher, indicating less change to the ML signal.

The better performance of both algorithms on the Non-Typical group for Turn may be due to the greater tendency for the typical group to adopt more aggressive turning strategies. Courtine et al. [31] found trunk roll during turning is greater at higher speeds, which would provide greater disruption to the signal. The 100% laterality determination accuracy for Non-Typical participants implies that the ML signal was not significantly disrupted. Observing the optical data showed that some typically functioning participants landed on their toes and/or pivoted strongly around one foot to complete a tight turning circle, while this was not the case for the non-typically functioning group. In contrast, the algorithm correctly identified 100% of the laterality for the non-typically functioning group. This may reflect the reticence for the non-typically functioning group to disrupt their walking pattern when planting their feet during the turn, though this should be investigated further.

Ultimately, gait stability assessment is used to benefit those with impaired function, so the fact that both algorithms performed relatively better for this group during tasks with horizontal head movements may improve its suitability for clinical use.

### 4.4. Limitations

This study had several limitations. Firstly, despite the fact that the cohort was diverse and included a significant number of participants with known movement disorders, none of these were highly severe. Including participants with higher levels of impairment would be a greater test of the algorithms’ ability to handle non-typical gait. Future work could also test the TP-EAR algorithm with very slow gait to see whether the peak corresponding to TC is still sufficiently prominent to be detected, as its amplitude has been shown to decrease with slower speeds [10].

Another limitation was the necessity of manual selection of the window length parameter when applying the SSA algorithm. To the authors’ best knowledge, this has not been discussed in the context of gait analysis using IMUs; however, there is existing research on finding the mathematically optimal window size for a given signal [38], which could provide an avenue for improving current SSA-based gait analysis implementations.

Additionally, while the scope of this study was to perform gait event detection for complex walking tasks used during functional assessments, it would be desirable to integrate this into in-the-wild monitoring. The set of activities performed was relatively contrived, so testing the algorithms in a less structured setting would allow assessment of their performance for more natural movements. Further to this, implementing the algorithm to run on-device and in real time would improve its transferability to a home-monitoring system. While running SSA is relatively computationally expensive since it relies on singular value decomposition, there exist efficient implementations which run on embedded devices [39].

Future work should look more thoroughly into the analysis of turning using ear-worn IMUs. It would be beneficial, for instance, to recreate the turning path and speed using the IMU signals, either to give context to the detected gait event or to look more deeply into the strategies adopted by varying participant groups. Analysis could also be extended to the sit-to-stand and stand-to-sit sections of the full timed up-and-go trial, which provide further movement-related challenges. While some work has been completed for this using glasses mounted with an IMU, this has yet to have been implemented using an ear-worn IMU [40].

## 5. Conclusions

This study investigated the performance of an ear-worn IMU system for detecting gait events during complex gait tests, comparing an existing (Diao) and a new (TP-EAR) algorithm. Both algorithms achieved highly accurate IC detection across the range of activities and participant groups, resulting in high accuracy in stride time calculation. TP-EAR outperformed the Diao algorithm for TC detection during walking with lateral head turns and while performing a turn, making it more suitable for determining stance and swing time parameters for movements which include lateral head movements. This work has direct applications in providing quantitative analysis of tasks which test gait stability in more flexible environments, thereby improving the monitoring of movement disorders. Further, the algorithms implemented have potential for future implementation of in-the-wild gait monitoring, where lateral head movements such as crossing a road or holding a conversation while walking provide everyday challenges to gait.

## Figures and Tables

**Figure 1 sensors-25-03629-f001:**
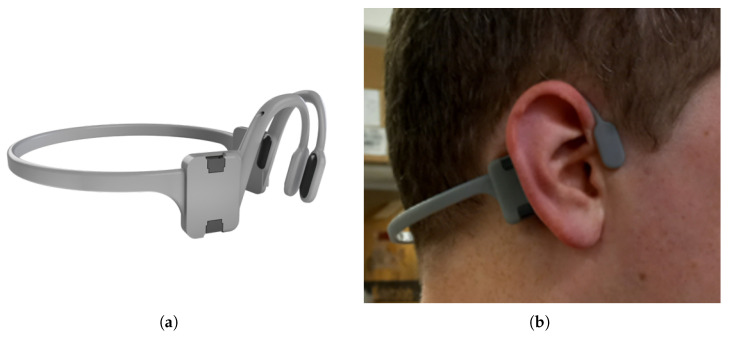
(**a**) Three-dimensional printed IMU headset used for data collection. (**b**) Headset being worn.

**Figure 3 sensors-25-03629-f003:**
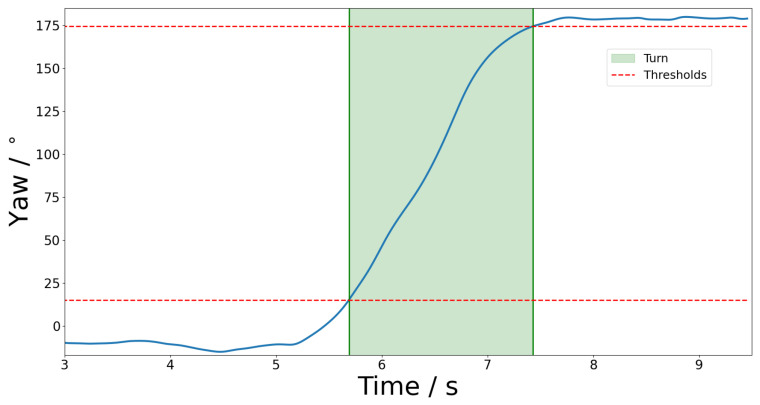
Segmentation of the Turn activity from the TUG trial using thresholding of the yaw signal.

**Figure 4 sensors-25-03629-f004:**
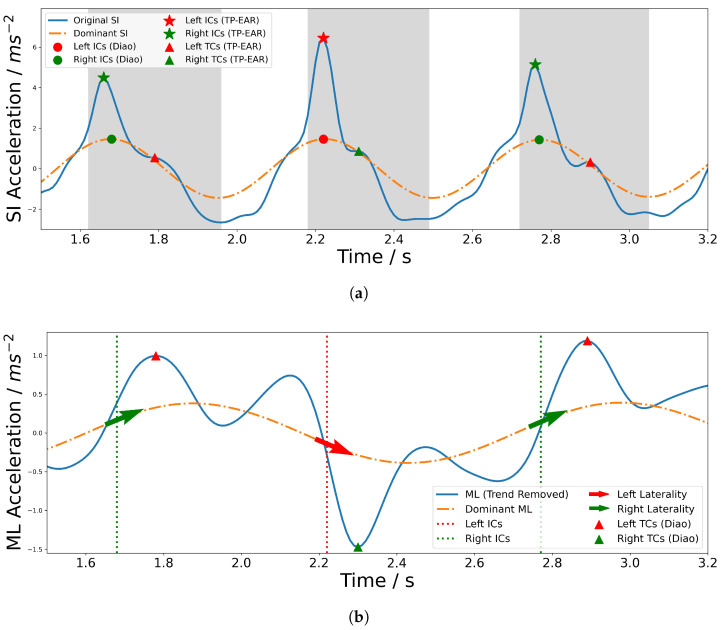
Demonstration of both algorithms applied on a gait cycle. (**a**) SI acceleration signal, where the grey areas illustrate the window defined by the TP-EAR algorithm for each cycle. (**b**) ML acceleration signal, where the coloured arrows show the IC laterality determined by the signed difference in the dominant ML signal.

**Figure 5 sensors-25-03629-f005:**
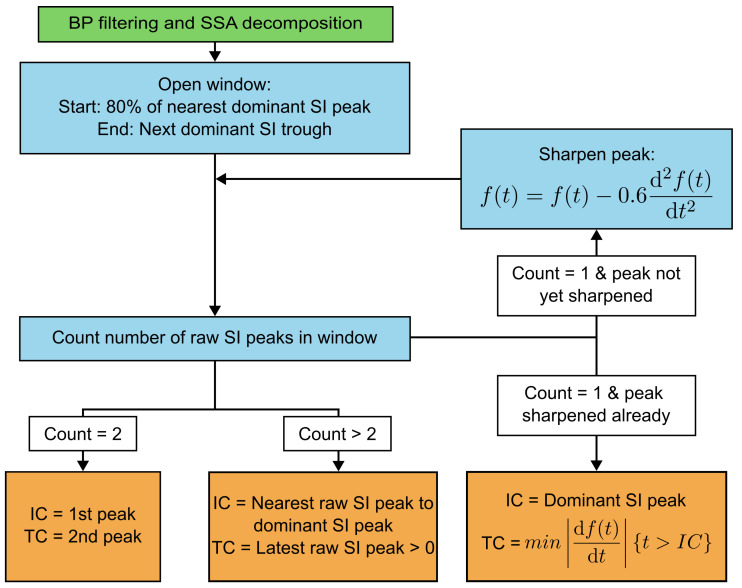
Process followed by TP-EAR to identify IC and TC events for a given walking bout. The green (preprocessing) step is applied to the whole bout, then the blue (processing) and orange (output) steps are applied to each dominant SI peak, which is an estimator to the IC location. f(t) is the windowed original SI acceleration signal. Laterality detection is then applied in the same way as for the Diao algorithm, demonstrated in Figure 4b.

**Figure 6 sensors-25-03629-f006:**
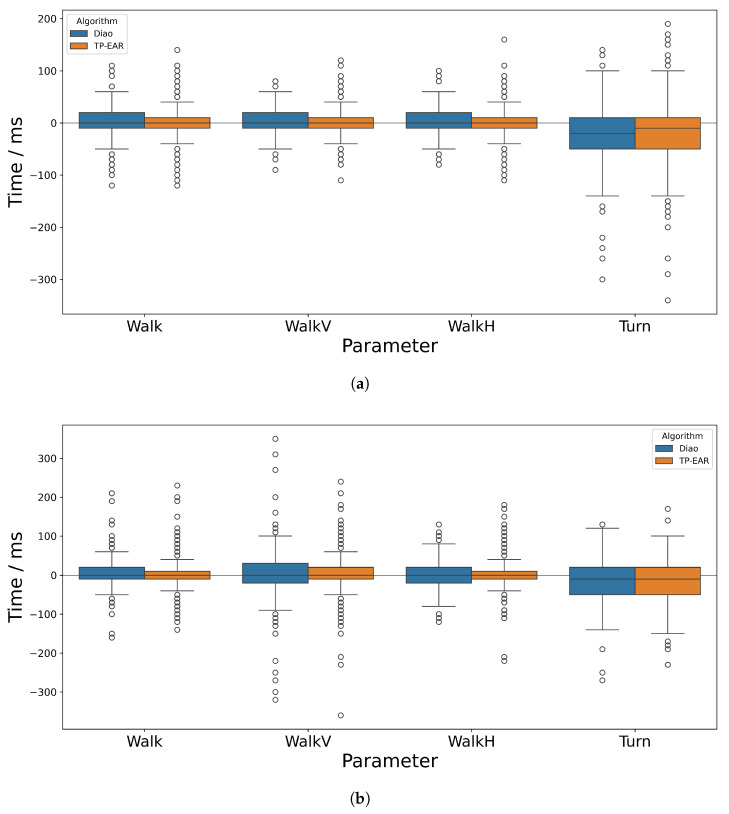
Time differences between IMU-derived and ground truth stride time for (**a**) Typical and (**b**) Non-Typical participant groups.

**Figure 7 sensors-25-03629-f007:**
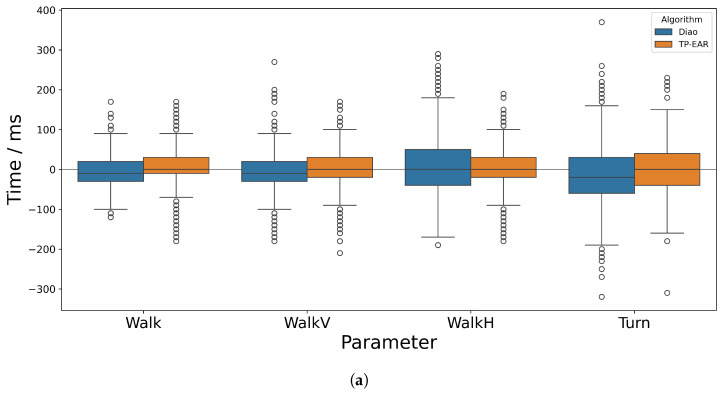
Time differences between IMU-derived and ground truth stance time for (**a**) Typical and (**b**) Non-Typical participant groups.

**Table 1 sensors-25-03629-t001:** Summary characteristics of the participants. Age, height and weight are reported as median (interquartile range).

Characteristic	Typical	Non-Typical
Sex (M/F)	18/32	10/8
Age (Years)	29.0 (15.0)	65.5 (22.0)
Height (m)	170.0 (14.8)	170.0 (16.0)
Weight (kg)	63.0 (16.0)	82.0 (17.8)

**Table 2 sensors-25-03629-t002:** Comparison of gait event detection sensitivity between the Diao and TP-EAR algorithms for each activity for typically functioning participants.

	Walk		WalkV		WalkH		Turn	
	Diao	TP-EAR	Diao	TP-EAR	Diao	TP-EAR	Diao	TP-EAR
IC (%)	99.9	99.9	99.9	99.9	99.9	99.8	99.4	99.4
TC (%)	100	99.9	99.7	100	97.1	99.8	91.4	98.7
Laterality (%)	100	99.8	99.7	99.4	82.8	83.0	94.6	93.7

**Table 3 sensors-25-03629-t003:** Comparison of gait event detection sensitivity between the Diao and TP-EAR algorithms for each activity for non-typically functioning participants.

	Walk		WalkV		WalkH		Turn	
	Diao	TP-EAR	Diao	TP-EAR	Diao	TP-EAR	Diao	TP-EAR
IC (%)	99.8	99.8	98.5	99.6	100	100	99.5	100
TC (%)	99.7	99.8	98.4	99.5	98.1	100	93.1	100
Laterality (%)	99.7	99.7	97.5	97.8	95.8	95.2	100	100

**Table 4 sensors-25-03629-t004:** Mean and standard deviation of difference in stride, stance and swing times between IMU and ground truth for typically functioning participants. TP stands for ‘temporal parameter’, AE denotes absolute error and SE denotes signed error. All values are reported in milliseconds.

	TP	Walk		WalkV		WalkH		Turn	
		Diao	TP-EAR	Diao	TP-EAR	Diao	TP-EAR	Diao	TP-EAR
AE (ms)	Stride	14±14	10±15	17±14	13±17	17±15	12±17	41±36	47±47
	Stance	26±22	28±29	33±33	36±34	60±57	37±34	64±58	54±46
	Swing	28±22	28±29	34±34	37±34	60±55	35±33	59±63	49±48
SE (ms)	Stride	4±20	0±18	3±22	1±21	3±23	1±21	−22±50	−20±63
	Stance	−4±34	7±39	−6±46	3±49	14±81	0±50	−12±86	−2±71
	Swing	8±35	−7±40	9±47	−2±50	−10±81	1±48	−8±86	−18±67

**Table 5 sensors-25-03629-t005:** Mean and standard deviation of difference in stride, stance and swing times between IMU and ground truth for non-typically functioning participants. TP stands for ‘temporal parameter’, AE denotes absolute error and SE denotes signed error. All values are reported in milliseconds.

	TP	Walk		WalkV		WalkH		Turn	
		Diao	TP-EAR	Diao	TP-EAR	Diao	TP-EAR	Diao	TP-EAR
AE (ms)	Stride	22±23	19±28	37±50	35±46	25±22	25±35	45±41	47±43
	Stance	47±42	52±42	64±67	73±69	76±63	55±52	75±54	68±60
	Swing	47±42	53±45	70±72	78±68	76±64	55±50	74±53	64±54
SE (ms)	Stride	5±31	2±34	2±62	3±58	3±34	2±43	−18±58	−17±61
	Stance	−22±59	10±66	−40±84	−26±97	−32±93	−7±75	−45±81	−37±83
	Swing	26±58	−9±69	46±89	29±100	34±93	8±74	28±87	21±81

## Data Availability

The datasets presented in this article are not readily available because the data is part of an ongoing study. Requests to access the datasets should be directed to T.F.

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
