# Peer review of "Detection of Gait Events Using Ear-Worn IMUs During Functional Movement Tasks"

_sensors, 2025, doi:10.3390/s25123629_

Round 1
Reviewer 1 Report
Comments and Suggestions for Authors
Manuscript: Detection of Gait Events Using Ear-worn IMUs During Functional Movement Tasks
Comments to authors:
The present study evaluates the use of ear-worn IMUs to detect gait events during gait tasks that included more than just straight walking. The paper is very well written. There are a few concerns, as listed below.
Introduction:
Line 46-47: ‘Also, the head is the body’s center of balance ….’ It is unclear what the authors mean by ‘center of balance’ and I am not sure if this could be considered a universal statement. Possibly provide references which discuss the head as the center of balance or just remove this specific part of the sentence.
Line 62: Should ‘lowest common subsequence’ not be ‘longest common subsequence’?
Materials and Methods:
Line 108: Why was the left ear only used? Please specify.
Line 137: ‘A small number of trials’ – it would be helpful to know how often this type of issue prevents good data. Could you possible state the frequency in parentheses – something like ‘X out of Y trials’ or ‘X% of trials’.
Line 137-138: Can you give any indication on why samples were ‘missed’ from the IMU?
Line 138: I assume missing data within the 0.3 s of IC causes issues with detecting the event, but please make this clear here, as in, explain why errors within 0.3 s is an issue?
Line 176-179: The authors mention heel rise and how this corresponds to a peak in the SI acceleration signal. However, heel rise does not occur at toe-off, but a short amount of time before actual toe-off? Can the authors please clarify how heel-rise, the SI acceleration peak, and the toe-off event relates?
Line 203: Here the authors mention both left and right ear but earlier (Line 108) states that only left ear data was used – please clarify?
Line 205-207: The fact that this window has to be changed based off length of trial is somewhat concerning in regard to the generalization of this method. Is this a limitation of the technique used here? If so, it should be addressed as a limitation in the discussion.
There is no mention of how ground-truth data was calculated. It only states that a OMC and RGB cameras were used together. Please add a brief description of how this was done
Results:
Line 244-245: Unless I misunderstand this, these values are not correct. For example, a change in TC Turn results from 91.4 (Diao) to 98.7 (TP-EAR) is stated as 7.1pp – is it not 7.3pp. There are similar issues for WalkH and WalkV. Please make sure of these numbers. This is similar for section 3.1.2 – please check the pp values.
Table 4 and 5 – please indicate units of measurement
Figure 5: These graphs show large ranges in differences, with some values up to 300 ms different from the ground truth. Please discuss these large outliers in your discussion section.
Discussion:
See comment about Figure 5 above.
Author Response
Comment 1: Line 46-47: ‘Also, the head is the body’s center of balance ….’ It is unclear what the authors mean by ‘center of balance’ and I am not sure if this could be considered a universal statement. Possibly provide references which discuss the head as the center of balance or just remove this specific part of the sentence.
Response 1: We acknowledge that this phrase was unclear - we meant that the vestibular system is located in the inner ear, which is a justification for why we place our sensor there for measuring functional movement. We have reworded this sentence on lines 46-47 to make this clearer.
Comment 2: Line 62: Should ‘lowest common subsequence’ not be ‘longest common subsequence’?
Response 2: Thank you for pointing this out, this is completely correct and has been changed on line 61.
Comment 3: Line 108: Why was the left ear only used? Please specify.
Response 3: Other works have used a single head or ear-worn sensor when performing gait event detection, so to maintain consistency we analysed a single ear-worn sensor, which we decided to be the left one. We have added the results obtained from analysing the right ear data, which were generally consistent with those obtained from the left ear, in Appendix A. We have added an explanation for this on lines 118-120.
Comment 4: Line 137: ‘A small number of trials’ – it would be helpful to know how often this type of issue prevents good data. Could you possible state the frequency in parentheses – something like ‘X out of Y trials’ or ‘X% of trials’.
Response 4: This happened in 45 out of 1208 trials, which is 3.7% of all trials. We have added this to line 151. To ensure the data quality was high, we excluded all the data from affected trials, even for the gait cycles within the trial which were unaffected.
Comment 5: Line 137-138: Can you give any indication on why samples were ‘missed’ from the IMU?
Response 5: The samples were missed due to lost packets during WiFi transmission of the collected data from the device to the laptop. This happened more frequently if the battery was running low, so if we observed this happening we would then replace the battery. We have added an explanation for this on lines 152-154.
Comment 6: Line 138: I assume missing data within the 0.3 s of IC causes issues with detecting the event, but please make this clear here, as in, explain why errors within 0.3 s is an issue?
Response 6: You are completely right, we have added a clearer justification for this in lines 152-155.
Comment 7: Line 176-179: The authors mention heel rise and how this corresponds to a peak in the SI acceleration signal. However, heel rise does not occur at toe-off, but a short amount of time before actual toe-off? Can the authors please clarify how heel-rise, the SI acceleration peak, and the toe-off event relates?
Response 7: Thank you for pointing this out. We had misinterpreted the relationship between heel rise and toe-off and incorrectly related this to the SI signal “indentation” mentioned by Zijlstra (2002) and Hwang (2018). The TP-EAR method relies on the observation that the indentation roughly coincides with toe-off and we have tested this during the study, with our results seeming to support this. However, the misguided attempt at a mechanistic explanation (i.e. heel-rise relating to this SI acceleration peak) has been removed.
Comment 8: Line 203: Here the authors mention both left and right ear but earlier (Line 108) states that only left ear data was used – please clarify?
Response 8: Please see our response to Comment 3.
Comment 9: Line 205-207: The fact that this window has to be changed based off length of trial is somewhat concerning in regard to the generalization of this method. Is this a limitation of the technique used here? If so, it should be addressed as a limitation in the discussion.
Response 9: Unfortunately, selection of window length is an inherent limitation of SSA which affects our methods and those produced by others, though has not been sufficiently acknowledged in other gait event detection methods which rely on SSA. We have added this to the Limitations section as you suggested on lines 515-519.
Comment 10: There is no mention of how ground-truth data was calculated. It only states that a OMC and RGB cameras were used together. Please add a brief description of how this was done
Response 10: Thank you for pointing this out. Primarily, the ground truth data was calculated by a manual observer using the RGB footage, however occasionally the RGB cameras were obstructed or the participant went out of their field of view. In these cases we used the OMC. We have added a brief description of how the RGB and OMC cameras were used to calculate the ground truth events on lines 127-129.
Comment 11: Line 244-245: Unless I misunderstand this, these values are not correct. For example, a change in TC Turn results from 91.4 (Diao) to 98.7 (TP-EAR) is stated as 7.1pp – is it not 7.3pp. There are similar issues for WalkH and WalkV. Please make sure of these numbers. This is similar for section 3.1.2 – please check the pp values.
Response 11: Thank you for pointing this out, this was a miscalculation on the authors’ part. We have checked and corrected this.
Comment 12: Table 4 and 5 – please indicate units of measurement
Response 12: We had included the units of measurement in the ‘TP’ row, however, we appreciate that this wasn’t sufficiently clear. We have now included the units of measurement in the table row headings and in the captions.
Comment 13: Figure 5: These graphs show large ranges in differences, with some values up to 300 ms different from the ground truth. Please discuss these large outliers in your discussion section.
Response 13: We agree that we should have discussed these outliers in the Discussion section. They occurred when a gait event was detected early then the following event was detected late (or vice-versa). If one or both of these events was wrongly estimated by a margin close to the cut-off (300ms), the calculated temporal parameter could therefore be out by more than 300ms. We have now added an explanation for this on lines 470-477.
Comment 14: See comment about Figure 5 above.
Response 14: See Response 13.
Reviewer 2 Report
Comments and Suggestions for Authors
The paper is well-written overall. However, I suggest addressing these minor points before publication:
1) Please define the full form of "TP-EAR" the first time it appears in the manuscript.
2) In the introduction, clearly compare the advantages and disadvantages of your proposed method with those of previously mentioned approaches.
Author Response
Comment 1: Please define the full form of "TP-EAR" the first time it appears in the manuscript.
Response 1: Thank you for pointing out that we had missed this – we have defined the full form of TP-EAR (Temporal Parameters from the EAR) on lines 106-107.
Comment 2: In the introduction, clearly compare the advantages and disadvantages of your proposed method with those of previously mentioned approaches.
Response 2: We have added greater discussion of the disadvantages, chiefly manual selection of the SSA window length parameter (requiring a minimum trial length) and lower prominence of the peak corresponding to TC, to the Limitations section (508-519). This complements the existing discussion of the method’s relatively high computational complexity (lines 520-528). We feel that we discussed the advantages of TP-EAR when we explained the algorithm in Section 2.4.2, specifically the fact that by relying on the SI signal we improve the robustness of TC detection during lateral head movements (lines 195-205). We believe that it is best to discuss the advantages and disadvantages of our method in these respective sections rather than move them to the Introduction.
Reviewer 3 Report
Comments and Suggestions for Authors
This paper uses ear-wore IMUs to detect gait on 68 people of different ages and types, and compares the new algorithms devised with the existing ones, achieving better accuracy in horizontal head rotations and turning movements.The paper shows rigorous in experimental design, experimental control, and data analysis, providing the possibility of walking gait detection in more complex scenarios. However, there are still several areas that could benefit from additional explanations.
- It is recommended that the four gait tasks be supplemented with diagrams to enhance the comprehension of the experiment.
- In the abstract, the author mentioned “Complex walking tasks such as turning or walking with head movements are frequently used to assess dysfunction in an individual’s vestibular, nervous and musculoskeletal systems”, however, in the paper, the description of why this research was carried out and its potential application areas and scenarios is scarce. It is recommended to increase the relevant discussion to improve the research value of the thesis.
- Some of the background colors in Figure 4 are suggested to be modified to make them clearer.
Author Response
Comment 1: It is recommended that the four gait tasks be supplemented with diagrams to enhance the comprehension of the experiment.
Response 1: Thank you for your suggestion - we have added Figure 1 which illustrates the four gait tasks on line 145.
Comment 2: In the abstract, the author mentioned “Complex walking tasks such as turning or walking with head movements are frequently used to assess dysfunction in an individual’s vestibular, nervous and musculoskeletal systems”, however, in the paper, the description of why this research was carried out and its potential application areas and scenarios is scarce. It is recommended to increase the relevant discussion to improve the research value of the thesis.
Response 2: We have added lines 96-100 in the Introduction to more explicitly describe the motivation for this research. Otherwise, we introduced common functional movement tests (Functional Gait Assessment and Berg Balance Scale) which include complex gait-related tasks in lines 19-28, and mention how turning and walking with head turns uncover movement-related deficiencies which do not appear during straight-line walking in lines 86-87. During the Discussion section, we suggested a possible application scenario of counting steps during a turn (which has been shown to differentiate between mildly impaired Parkinson’s Disease patients and age-matched controls) on lines 407-409. We also link back to the application of our work in gait stability tests and speculate on future implementation for in-the-wild gait monitoring in the Conclusion (lines 541-546). Therefore, we believe we have now included sufficient discussion of application areas.
Comment 3: Some of the background colors in Figure 4 are suggested to be modified to make them clearer.
Response 3: Thank you for pointing this out – we have updated the Figure (now Figure 5 on line 206) to have lighter background colours on black text, which we believe makes the content clearer.
Reviewer 4 Report
Comments and Suggestions for Authors
Review comments:
1, The introduction provides a comprehensive background on gait analysis using wearable inertial measurement units (IMUs) and specifically focuses on ear-worn IMUs. It discusses the advantages of using wearable sensors over traditional methods and highlights the relevance of gait event detection during complex walking tasks. The references cited are relevant and support the introduction well. However, a few more recent references on gait analysis using ear-worn IMUs could strengthen the literature review.
2, The research design is appropriate. The study compares the performance of an existing gait event detection algorithm with a new algorithm using data collected from a diverse group of participants, including those with movement disorders. The inclusion of complex walking tasks like turning and walking with head movements makes the study more clinically relevant.
3, The methods are adequately described. The data collection process, the IMU headset design, and the gait event detection algorithms are clearly explained. The definitions of gait events and the evaluation metrics used are also well defined.
4, The results are clearly presented. The sensitivity, laterality accuracy, and temporal parameter errors are reported for both algorithms across different activities and participant groups. The tables and figures complement the text and make the results easy to follow.
5, The conclusions are well supported by the results. The new algorithm (TP-EAR) shows improved performance for terminal contact detection during complex walking tasks, which is in line with the findings reported. The discussion also provides a good interpretation of the results and highlights the clinical implications.
6, The figures and tables are clear and well-presented. They are labeled appropriately and the legends provide sufficient details. The use of color coding in the figures helps in distinguishing between different algorithms and participant groups.
Here are some suggestions for improving the manuscript:
- While the methods section provides a good overview of the data collection and analysis, more details on the preprocessing steps and algorithm implementation would enhance reproducibility.
- The use of 'earables' could be defined when first introduced to make the manuscript more accessible to a general audience.
- Consider adding a few more recent references on gait analysis using ear-worn IMUs to strengthen the literature review."
The manuscript presents an interesting study with promising results. With some minor revisions to address the points mentioned above, it could be suitable for publication.
Author Response
Comment 1: While the methods section provides a good overview of the data collection and analysis, more details on the preprocessing steps and algorithm implementation would enhance reproducibility.
Response 1: Thank you for your suggestion – we have added a dedicated Data Preprocessing section in the methods section (Section 2.2) which begins on line 156. Regarding the algorithm implementation, we feel that this has already been explained in sufficient depth in the Gait Event Detection section (Section 2.4). Particularly, for TP-EAR we have included a dedicated subsection (Section 2.4.2) with a written description of the algorithm accompanied by Figure 5 which outlines the algorithm as a flow chart. Further, both the TP-EAR and Diao algorithms are demonstrated in Figure 4. We have also made the code available on Github as mentioned on line 181.
Comment 2: The use of 'earables' could be defined when first introduced to make the manuscript more accessible to a general audience.
Response 2: We believe we defined the term 'earables' in the phrase: “IMU sensors placed at the ear, often referred to as 'earables'… ”. Since the term is not used again in the manuscript, we don’t feel that a more detailed definition is required.
Comment 3: Consider adding a few more recent references on gait analysis using ear-worn IMUs to strengthen the literature review.
Response 3: We have added two more recent references concerning gait analysis using ear-worn IMUs on Lines 73-79. These are by Decker et al. (2024) and Jung et al. (2023) which correspond to [23] and [24] in the References section.
Round 2
Reviewer 1 Report
Comments and Suggestions for Authors
Manuscript: Detection of Gait Events Using Ear-worn IMUs During Functional Movement Tasks
Comments to authors:
I want to thank the authors for all revisions made regarding the comment and suggested edits. Everything has been clearly addressed. There are a two very minor issues remaining (see below).
Introduction:
Line 61: I am confused why this was not corrected, and assume the authors just forgot? Should ‘lowest common subsequence’ not be ‘longest common subsequence’?
Materials and Methods:
Line 205: ‘Figure??’ Please add figure number.
Author Response
Comment 1: Line 61: I am confused why this was not corrected, and assume the authors just forgot? Should ‘lowest common subsequence’ not be ‘longest common subsequence’?
Response 1: Thank you, this is correct and was another oversight by us. This has now been corrected.
Comment 2: Line 205: ‘Figure??’ Please add figure number.
Response 2: This seems to have resulted from two of our figures (Figures 1 and 5) having been commented out after submission (meaning their reference label was also missing). We are unsure why this happened but potentially it was due to these being .svg files, so we have updated the manuscript to include these as .png files instead.